# Design of an Optical Probe to Monitor Vaginal Hemodynamics during Sexual Arousal

**DOI:** 10.3390/s19092129

**Published:** 2019-05-08

**Authors:** Hyeryun Jeong, Myeongsu Seong, Hyun-Suk Lee, Kwangsung Park, Sucbei Moon, Jae Gwan Kim

**Affiliations:** 1Department of Biomedical Science and Engineering, Gwangju Institute of Science and Technology (GIST), Gwangju 61005, Korea; ryun@gist.ac.kr (H.J.); myeongsuseong@sjtu.edu.cn (M.S.); 2Department of Urology, Chonnam National University Medical School, Gwangju 61186, Korea; sys23@hanmail.net (H.-S.L.); uropark@gmail.com (K.P.); 3Department of Physics, Kookmin University, Seoul 02707, Korea; moons@kookmin.ac.kr

**Keywords:** female sexual dysfunction, apomorphine, hemodynamics, vaginal wall, sexual arousal, temperature

## Abstract

An optical probe was developed to measure the change of oxy-hemoglobin (OHb), deoxy- hemoglobin (RHb), and total hemoglobin (THb) along with temperature from the vaginal wall of female rats. Apomorphine (APO, 80 μg/kg) was administered to elicit sexual arousal in female Sprague Dawley rats (SD, 180–200 g). The behavior changes caused by APO administration were checked before monitoring vaginal responses. The changes of oxy-, deoxy-, and total hemoglobin concentration and the temperature from the vaginal wall were monitored before, during, and after APO administration. Animals were under anesthesia during the measurement. After APO administration, the concentration of OHb (55 ± 29 μM/DPF), RHb (33 ± 25 μM/DPF), and THb (83 ± 59 μM/DPF) in the vaginal wall increased in a few min, while saline administration did not cause any significant change. In case of the vaginal temperature change, APO decreased the temperature slightly in the vaginal wall while saline administration did not show any temperature change in the vaginal wall. As the outcomes demonstrated, the developed probe can detect hemodynamic and temperature variation in the vaginal wall. The hemodynamic information acquired by the probe can be utilized to establish an objective and accurate standard of female sexual disorders.

## 1. Introduction

Aging leads to the decline of estrogen levels in women, which can result in female sexual dysfunction (FSD) [1]. With diminished ovarian estrogen, the vaginal wall becomes thinner and physical characteristics, such as elasticity, expandability, and flexibility, worsen. Postmenopausal women, in particular, experience discomfort when it comes to having intercourse because menopause entails low desire or interest, diminished arousal, orgasmic difficulties, and dyspareunia [2,3]. Female sexual dysfunction is widely reported to be prevalent all over the world; there is a growing interest in it, and a few studies have been carried out on the matter [4,5].

Several tools have been applied to diagnose female sexual dysfunction, such as pH lubrication measurement [6], functional MRI (fMRI) [7], and laser Doppler flowmetry (LDF) [8]. Among these tools, pH measurement in the vaginal wall is a common and simple means of evaluation. It is possible to collect vaginal fluid and find out the hormone associated with vaginal pH, but the sample is easily contaminated with urine and time-dependent. Monitoring temperature in the vaginal wall during sexual arousal is also one of the critical parameters to study female sexual dysfunction [9]. There are several studies that have measured the temperature of the body or genital organ during sexual arousal under different sexual stimulation schemes. Functional MRI has the advantage of clarifying brain function correlated with sexual arousal and response, but it is high cost and generates loud noises [10].

Laser Doppler flowmetry (LDF) is currently used in clinics to diagnose female sexual dysfunction since sexual arousal results in physiological changes, such as the increase of blood flow to the lower abdomen and genital [8]. The vaginal wall has a bunch of small vessels, including capillaries. Sexual arousal induces expansion of such small blood vessels and of the vascular layer of the vaginal wall, resulting in an increase of blood flow [4]. However, given the vascular structure of the vaginal wall, LDF would not reflect the vaginal wall’s hemodynamic response since it mainly targets blood flows in arteries. Therefore, a tool needs to be developed to acquire signals from microvessels in the vaginal wall that will more accurately represent the response to sexual arousal.

Near-infrared spectroscopy (NIRS) utilizes near-infrared wavelength light to provide hemodynamic information, such as relative change of oxy-, deoxy-, and total hemoglobin concentration. NIRS can be a useful diagnostic tool to monitor hemodynamics in an FSD study [11]. It is also known to be sensitive to microvessels, which are widely distributed in the vaginal wall, and thus it fits well for the purpose of monitoring microvascular hemodynamics in the vaginal wall compared to LDF [12].

In this study, we designed an optical probe based on NIRS to monitor vaginal hemodynamics along with temperature during sexual arousal induced by apomorphine administration. Our aim was to find out whether the optical probe with NIRS could be a tool to assess female sexual dysfunction.

## 2. Materials and Methods

A probe has been designed considering the shape and size of the animal’s vagina after dissection. Figure 1 shows the schematic of the designed probe. A whole cylindrical stainless-steel tube with 3 mm of diameter was used as a probe sheath. Three holes were made to place optical fibers for light delivery and detection. On the probe, one source fiber and one detector were placed having a separation of 1.5 mm. Optical fibers (FG105UCA-CUSTOM, Thorlabs, USA) with 105/125 μm of core/cladding diameter were used for both illumination and detection of the light. A microthermistor (10k Ohms, TH10K, Thorlabs, USA) was placed at the side of the probe to measure the vaginal wall temperature (Figure 1).

Modified Beer–Lambert law (MBLL) was utilized to obtain the change of oxy-hemoglobin (OHb), deoxy-hemoglobin (RHb), and total hemoglobin (THb) concentration [13]. We assume that the main chromophores in the near-infrared wavelengths are OHb and RHb in biological tissue. The change in optical density (Δ*OD*) at given wavelengths can be represented with the change of OHb (Δ*OHb*) and RHb (Δ*RHb*) by
(1)$$\left[\begin{array}{c} \begin{array}{c} ∆OD^{730}\\ ∆OD^{750} \end{array}\\ ∆OD^{800}\\ ∆OD^{830}\\ ∆OD^{850} \end{array}\right]=\left[\begin{array}{cc} \varepsilon_{OHb}^{\lambda 1} & \varepsilon_{RHb}^{\lambda 1}\\ \begin{array}{c} \varepsilon_{OHb}^{\lambda 2}\\ \begin{array}{c} \varepsilon_{OHb}^{\lambda 3}\\ \begin{array}{c} \varepsilon_{OHb}^{\lambda 4}\\ \varepsilon_{OHb}^{\lambda 5} \end{array} \end{array} \end{array} & \begin{array}{c} \varepsilon_{RHb}^{\lambda 2}\\ \begin{array}{c} \varepsilon_{RHb}^{\lambda 3}\\ \begin{array}{c} \varepsilon_{RHb}^{\lambda 4}\\ \varepsilon_{RHb}^{\lambda 5} \end{array} \end{array} \end{array} \end{array}\right]\left[\begin{array}{c} ∆OHb\\ ∆RHb \end{array}\right]L$$
where Δ*OD* = *OD_Transient* − *OD_Baseline*, *ε* (extinction coefficient) of OHb and RHb at five wavelengths (730, 750, 800, 830, and 850 nm), and *L* is an optical path length between light source and detector. In a highly scattering medium, the optical path length of a detected photon is approximated as *L* = *d* × *DPF* (*d*: source and detector separation, *DPF*: differential path length factor). The value of Δ*OHb*, Δ*RHb*, and Δ*THb* can be obtained with the assumption that scattering is constant during the experiment (Equation (2)) and Δ*THb* is a summation of Δ*OHb* and Δ*RHb* (Equation (3)).
(2)$$\left[\begin{array}{c} ∆OHb\\ ∆RHb \end{array}\right]=\;\frac{1}{d.DPF}{\left[\begin{array}{cc} \varepsilon_{OHb}^{730} & \varepsilon_{RHb}^{730}\\ \begin{array}{c} \varepsilon_{OHb}^{750}\\ \begin{array}{c} \varepsilon_{OHb}^{800}\\ \begin{array}{c} \varepsilon_{OHb}^{830}\\ \varepsilon_{OHb}^{850} \end{array} \end{array} \end{array} & \begin{array}{c} \varepsilon_{RHb}^{750}\\ \begin{array}{c} \varepsilon_{RHb}^{800}\\ \begin{array}{c} \varepsilon_{RHb}^{830}\\ \varepsilon_{RHb}^{850} \end{array} \end{array} \end{array} \end{array}\right]}^{-1}\left[\begin{array}{c} \begin{array}{c} ∆OD^{730}\\ ∆OD^{750} \end{array}\\ ∆OD^{800}\\ ∆OD^{830}\\ ∆OD^{850} \end{array}\right]$$
Δ*THb* = Δ*OHb* + Δ*RHb*(3)

For the experimental setup, a probe was gently inserted into the vagina of each animal. The probe was connected to the NIRS system, which consists of a broadband light source (HL-2000-HP, Ocean Optics, USA) and an NIR spectrometer (USB4000, Ocean Optics, USA). A thermistor on the other side of the probe was connected to a data acquisition system (USB-6259 BNC, National Instruments, USA) for the acquisition of analog voltage signals (Figure 2). The collected analog voltage signals of the thermistor were converted to temperature using conversion equations given by the manufacturer.

Animals were housed in a polycarbonate cage with a stainless lid under a 12/12 h light/dark cycle. Female Sprague Dawley rats (180 to 220 g, *n* = 8) were used in the study. Each animal was measured twice to minimize group variability, first, with saline administration (control group) and, second, with apomorphine hydrochloride hemihydrate (APO) administration (APO group). The animals could access water and foods ad libitum. APO (A4393, Sigma-Aldrich, USA) was administered subcutaneously to each animal with a dose of 80 μg/kg. For comparison, the same volume of saline was administered to the animals. A warm water pad was placed under the rats to prevent hypothermia during the measurement. The Institutional Animal Care and Use Committee of the Gwangju Institute of Science and Technology approved this study (April 12, 2017).

A respiratory challenge has been applied to the animals to verify the developed probe. Figure 3 shows the protocol for the respiratory challenge. Each animal was anesthetized using 1–1.5% isoflurane during the measurement. A gas mixer connected with an isoflurane vaporizer was used to control the constituents of supplied gas for the animal. The developed probe was positioned inside the vagina of each rat during the respiratory challenge. Air (21% oxygen + 79% nitrogen), hypoxic gas (16% oxygen + 84% nitrogen), and air again were supplied for the first 20 min, 15 min, and 20 min, respectively, while monitoring the vaginal hemodynamic signals (Figure 3).

Figure 4 shows the protocol for the APO experiment. The animal was first placed in an anesthesia chamber for induction for 3 min supplying isoflurane (3% to 5%). Once the animal was anesthetized, ketamine (50 mg/kg) was administered intraperitoneally to maintain the anesthesia, and there was no additional supply of isoflurane during the measurement. The developed probe was positioned inside the vagina of each rat during sexual stimulation with the administration of APO. After the first 5 min of baseline signal measurement, APO was administered, and vaginal hemodynamics was continuously monitored for another 50 min. During the measurement, 50% oxygen and 50% nitrogen were supplied to maintain the level of arterial oxygen saturation (~98%). 

## 3. Results

After administration of APO, most animals showed a patterned behavior response, including a startled and licking response, yawning, and bleary eyes that lasted for 30–60 min, and a few animals also showed specific sexual-appetite-related behavior, such as a genital grooming response, sniffing, extension of the neck, and movement of the head towards their genitals.

Figure 5 shows the representative results of the respiratory challenge test. For the first 20 min of air inhalation, there were no significant changes in OHb, RHb, and THb. OHb concentration decreased, and RHb concentration increased when gas was switched to hypoxic gas. The signals were returning towards the baseline step after air inhalation.

Figure 6a,b demonstrate concentration changes of OHb, RHb, and THb from the control group and the APO administration group, respectively, obtained from eight rats. All the data from each group were averaged and compared with the control group. Figure 6a shows that there was no significant change of OHb, RHb, THb, and temperature when saline was administered. On the other hand, Figure 6b shows that APO administration induced an increase of the concentration of OHb (0.055 ± 0.029 mM/DPF), RHb (0.033 ± 0.025 mM/DPF), and THb (0.083 ± 0.059 mM/DPF). The vaginal temperature was initially 37.3 degrees. It decreased 0.3 degrees right after APO administration and dropped gradually to 36.7 degrees later on.

Figure 7 represents the delayed response time after the administration of APO for OHb, RHb, and THb. The response time was the time interval between the beginning of APO administration and the summit of the concentration change for each chromophore. The response time of OHb and THb was the same, while RHb showed a delayed response time. APO was administered via the subcutaneous route (s.c.), which is a safe and convenient method when a small volume is applied but which produces a slower response time than the intravenous (i.v.) route. Average time differences between baseline and peak of the concentration change for OHb, RHb, and THb are 17.28 min, 18.94 min, and 16.90 min, respectively.

## 4. Conclusions and Discussion

In this study, a diffuse optical probe was fabricated for investigating the hemodynamic response and temperature change of the vaginal wall of rats. APO was administered as a sexual stimulant, and animal behavior and hemodynamic changes were monitored following APO administration. APO administration induced significant behavior changes in non-anesthetized rats, such as a startled and licking response, yawning, and bleary eyes. In contrast to the control group, APO caused not only behavior changes but also dramatic hemodynamic changes and a temperature decrease in the vaginal wall during the measurement. Therefore, these preliminary results indicate that our system is capable of monitoring hemodynamic responses, including blood volume and tissue oxygenation, during sexual stimulation.

Interestingly, a slight decrease in temperature was observed in the APO group while no temperature change was observed in the control group. A temperature change in the vaginal wall is evidence of sexual arousal, and, therefore, it is also a potential indicator of female sexual dysfunction. It was expected that the vaginal temperature would increase with the response to the APO administration, but it did not. It has been shown that the temperature can change in response to pelvic nerve stimulation, APO administration, or medial preoptic area stimulation, and it can be dependent on particular areas, such as labia minora, labia majora, or vaginal wall [14,15,16]. In addition, the temperature response to APO is reported to be dose-dependent [17], making it difficult to use temperature as a parameter to monitor sexual arousal in animal models.

The hemodynamic results from this study can be explained by the process of proper vaginal function after APO administration. APO is a dopaminergic agent released in the striatum and nucleus accumbens. What is more, it is a central dopamine receptor that regulates primarily through parasympathetic oxytocinergic nerve fibers, and it can increase arterial blood supply [18]. Thereby, the APO administration enhances blood supply to the vaginal wall and genital organs. Therefore, the hemodynamic information acquired by the optical probe can be utilized to offer quantitative information about female sexual disorders (FSD).

To minimize its effect on the hemodynamic signal, pressure could be reduced by measuring the vaginal size during the dissection of rats (3.13 mm ± 0.34 mm) before the design of the optical probe. The diameter of the probe was set to 3 mm so that the probe did not push the vagina wall and at the same time the vagina was not too loose to hold the probe inside it. To do that, vaginal information was measured by dissection of rats (N = 3). The reflectance intensity value from the probe was also controlled carefully when the probe was placed inside of the vagina because the reflectance intensity varied depending on how much the optical probe pushed the vagina wall. We tried to keep a similar intensity of reflectance at 700 nm for all the subjects. Therefore, we believe that the pressure effect on the hemodynamic signal was minimized in this study.

In the future, we will investigate vaginal dryness, which is a serious FSD, using the developed probe. The same experimental protocol will be applied to a group with vaginal dryness that have undergone ovariectomy to find out the difference in hemodynamic response compared to the normal group. The future study will be a strong indicator to diagnose female sexual dysfunction since we believe that the developed diffuse optical probe is translational to clinics by adjusting the probe size.

## Figures and Tables

**Figure 1 sensors-19-02129-f001:**
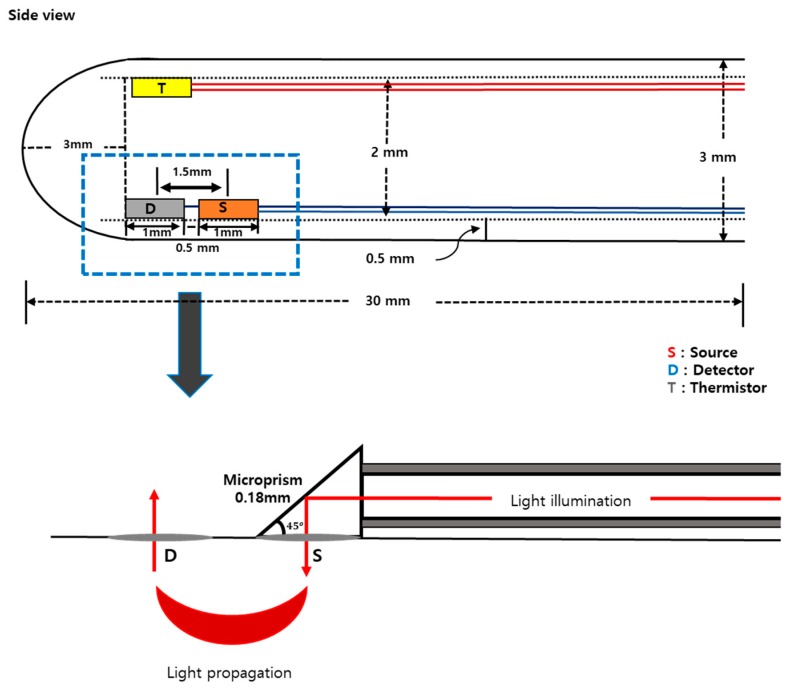
A schematic diagram of the probe’s design.

**Figure 2 sensors-19-02129-f002:**
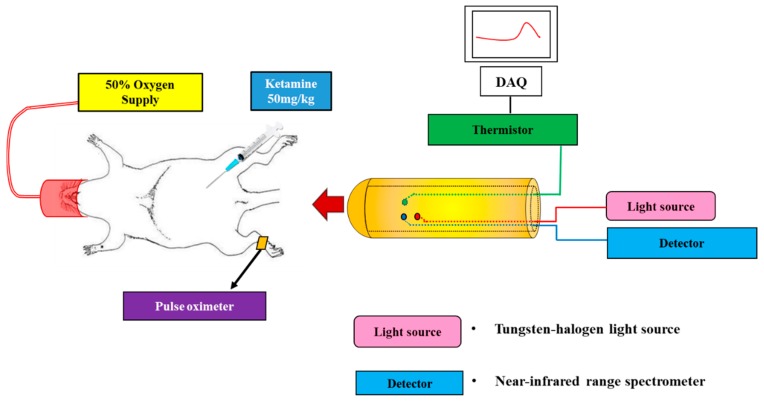
Experimental setup.

**Figure 3 sensors-19-02129-f003:**
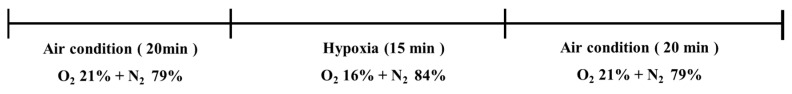
Respiratory challenge protocol.

**Figure 4 sensors-19-02129-f004:**
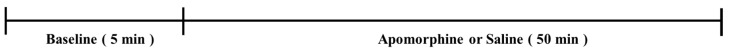
Apomorphine (APO) or Saline-induced protocol.

**Figure 5 sensors-19-02129-f005:**
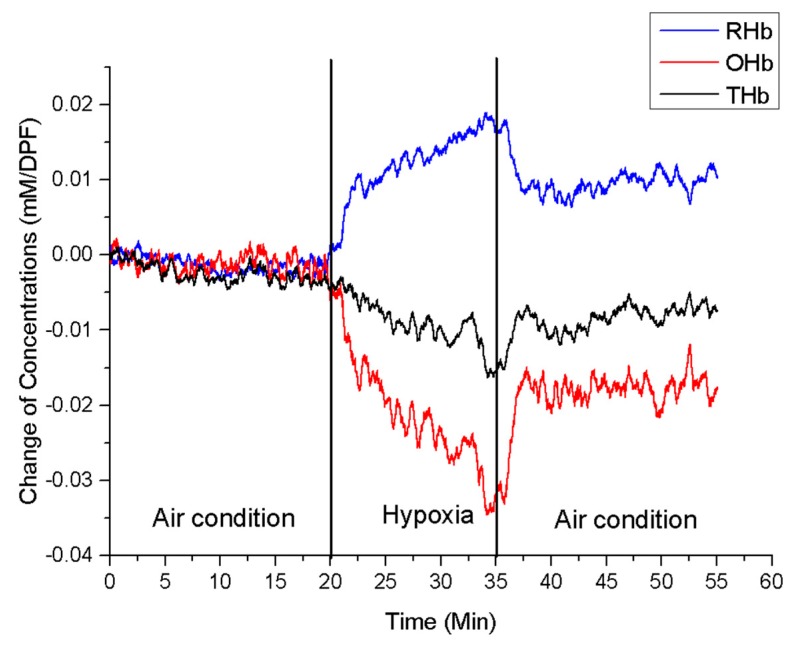
Representative result of the respiratory challenge test. (* RHb: deoxy-hemoglobin, OHb: oxy-hemoglobin, THb: total hemoglobin, mM: millimole, DPF: Differential path length factor).

**Figure 6 sensors-19-02129-f006:**
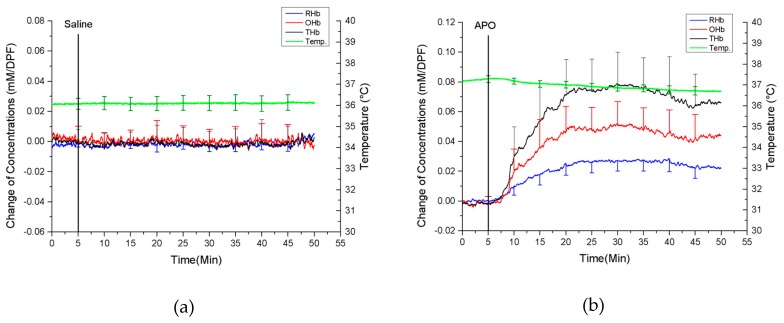
The average hemodynamic changes from all rats (**a**) Control group (**b**) APO group. (* RHb: deoxy-hemoglobin, OHb: oxy-hemoglobin, THb: total hemoglobin, mM: millimole, DPF: Differential path length factor).

**Figure 7 sensors-19-02129-f007:**
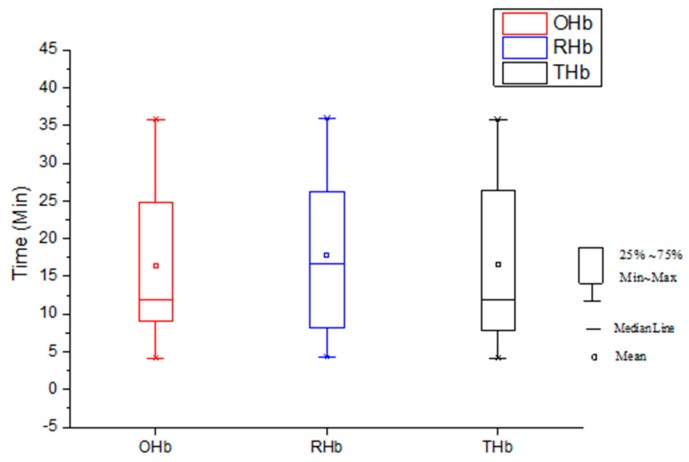
Drug (APO) Response Time. (* RHb: deoxy-hemoglobin, OHb: oxy-hemoglobin, THb: total hemoglobin).

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
