# Peer review of "Design of an Optical Probe to Monitor Vaginal Hemodynamics during Sexual Arousal"

_sensors, 2019, doi:10.3390/s19092129_

Round 1

Reviewer 1 Report

This work looks interesting and promising. I recommend the paper for publishing in this journal after the some concerns have been addressed. 

1. The authors need to explain why the source-detector separation of 1.5 and 2 mm is chosen. What sampling volume (or the depth of radiation penetration) is typical for such a design of the probe?

2. For the reader's convenience, it would be useful to give a brief description of the MBLL method in the text of the article.

3. How did authors control the pressure of the probe on the vaginal wall? This is an important parameter that could affect the measurement results (see e.g.: A. Popov et al. “Influence of probe pressure on diffuse reflectance spectra of human skin measured in vivo”, JBO, 2017; E. Zherebtsov et al. “The influence of local pressure on evaluation parameters of skin blood perfusion and fluorescence”, Proc. SPIE, 2017; I. Mizeva et al. “Optical probe pressure effects on cutaneous blood flow”, Clinical hemorheology and microcirculation, 2019). It should be taken into account that the parameters of the vagina of each individual animal are variable and the pressure of the probe could be different.

Author Response

We would like to thank for reviewing our manuscript and providing valuable comments and important suggestions so that the manuscript can be further improved. Our response to the comments comes below.

[Major changes]

Point 1: The authors need to explain why the source-detector separation of 1.5 and 2 mm is chosen. What sampling volume (or the depth of radiation penetration) is typical for such a design of the probe?

Response 1

The probe was designed considering the size and shape of the vagina of the female rat. In general, the penetration depth is half of the distance between the light source and detector in the near-infrared range, and therefore, 1.5 and 2mm source-detector separation will detect hemodynamic signal from the depth of around 0.75~1mm. The thickness of the vaginal wall under anesthesia state in the rat (~200 g) was 1.34 ± 0.13 mm (N=3) which is thicker than the range of detection by having 1.5 and 2mm source-detector separation. However, we collected hemodynamic signal only from 1.5 mm source-detector separation because 2 mm separation may probe the signal beyond the vaginal wall.

Since we only collected data from 1.5 mm separation, Figure 2 has been redrawn after the removal of detector 2 mm away from the light source fiber.

Point 2: For the reader's convenience, it would be useful to give a brief description of the MBLL method in the text of the article.

Response 2

A brief MBLL has been described in the manuscript as below.

Page. 2, line 77 – Page. 3, line 93

Point 3: How did authors control the pressure of the probe on the vaginal wall? This is an important parameter that could affect the measurement results. It should be taken into account that the parameters of the vagina of each individual animal are variable and the pressure of the probe could be different.

Response 3

Thank you for the very important comment. Pressure can certainly affect the hemodynamic signal since it is calculated from the light attenuation in which is affected by the pressure between the probe and tissue. We added the following paragraph in the discussion section.

Page. 7, line 200 - Page. 7, line 209

To minimize the pressure effect on the hemodynamic signal, the vaginal size was measured by dissection of rats (3.13 mm ± 0.34 mm) before the design of the optical probe. The diameter of the probe was set to 3 mm so that the probe does not push the vagina wall and at the same time the vagina is not too loose to place the probe inside the vagina. The reflectance intensity value from the probe was also controlled carefully when the probe was placed inside of the vagina because the reflectance intensity varied depending on how much the optical probe pushes the vagina wall. We tried to keep a similar intensity of reflectance at 700 nm for all the subjects. Therefore, we believe that the pressure effect on the hemodynamic signal was minimized in this study.”

Reviewer 2 Report

In the paper entitled, “Vaginal Hemodynamic Change During Sexual Arousal in a Rat Model Using a Diffuse Optical Probe,” the authors presented a new DOS tool for measuring vaginal hemodynamics, i.e. oxy-, deoxy- and total hemoglobin,  in rats. Along with the hemodynamics, probe is able to register temperature changes. The increasing of oxy-, deoxy- and total hemoglobin in rats vagina (N=8) has been successfully measured during sexual arousal induced by Apomorphine injection. Administration of saline solution and respiratory challenge protocol were applied in order to test the probe. The authors believe that such probe adjusted by the size can be translated into the clinic to diagnose female sexual dysfunction.  The paper is of interest to the biophotonics society and gynecologists, nevertheless, the paper should pass through major revision process before publication by the following reasons:   

1. The motivation of the study is not enough strong (“growing interest in female sexual disorder” is not enough for motivation). The importance of instrumentation measurements of female sexual disorders should be strengthened.

2. The DOS probe is not enough described:

2.1. Figure 1 shows that the fiber tip surfaces are perpendicular to the probe surface. In this case, how do they illuminate the surface and collect the scattered light? Are there any mirrors on the fiber tips or the Figure 1 gives wrong representation about the fiber tips orientation? This issue should be clarified.

2.2. DOS measurements are usually works ok for relative measurements, like detection of an increase or decrease of the oxy- and deoxy- hemoglobin in time, but it is hard to obtain quantitative information. Quantitative data in DOS is very sensitive to the probe geometry and investigated tissue and a number of phantom studies and numerical simulations are usually applied in order to prove the obtained numbers. The authors give quantitative information on the blood content in mM/DPF (by the way, what is the “DPF”?), but no data on the probe calibration have been provided.

If the data presented is only relative, the authors should provide evidence that the relative data will be enough to measure the female sexual disorders in future.

Minor changes.

1. In the title “rat model” sounds like the final aim of this study is to investigate sexual arousal in rats. Please, modify the title.

2. “DPF” abbreviation on Figures 5,6 should be decrypted.

Author Response

Comments to the Author

We would like to thank for reviewing our manuscript and providing valuable comments and important suggestions so that the manuscript can be further improved. Our response to the comments comes below.

[Major changes]

Point 1: The motivation of the study is not enough strong (“growing interest in female sexual disorder” is not enough for motivation). The importance of instrumentation measurements of female sexual disorders should be strengthened.

Response 1

In order to strengthen the importance of instrumentation, several diagnostic tools for female sexual dysfunction (FSD) has been added, described and revised in the introduction.

Page. 1, line 39 – Page. 2, line 47

Even though Laser Doppler flowmetry (LDF) has been extensively used as a diagnostic tool of female sexual dysfunction, it has drawbacks. Therefore, we performed this study to find if this optical tool can be used as a diagnostic tool of female sexual dysfunction. We mentioned the importance of this study in the manuscript, and it was highlighted with yellow.

Page. 2, line 48 – Page. 2, line 55

Point 2.1: Figure 1 shows that the fiber tip surfaces are perpendicular to the probe surface. In this case, how do they illuminate the surface and collect the scattered light? Are there any mirrors on the fiber tips or the Figure 1 gives wrong representation about the fiber tips orientation? This issue should be clarified.

Response 2.1

A microprism (0.18mm, Thorlab) was glued at the end of the fiber so that light could be transmitted and detected perpendicular to the tube. We revised Figure 1 by adding an inlet figure to show the detail at the end of the source and detector fiber.

Point 2.2: DOS measurements are usually works ok for relative measurements, like detection of an increase or decrease of the oxy- and deoxy- hemoglobin in time, but it is hard to obtain quantitative information. Quantitative data in DOS is very sensitive to the probe geometry and investigated tissue and a number of phantom studies and numerical simulations are usually applied in order to prove the obtained numbers. The authors give quantitative information on the blood content in mM/DPF (by the way, what is the “DPF”?), but no data on the probe calibration have been provided.

If the data presented is only relative, the authors should provide evidence that the relative data will be enough to measure the female sexual disorders in future.

Response 2.2

1. A continuous wave near-infrared spectroscopy (CW-NIRS) system was used in this study, and it provides only the relative changes in chromophore concentration.

2. DPF is the abbreviation of differential pathlength factor which is the ratio between the mean pathlength of photons and direct distance between a light source and a detector. Therefore, it increases as the scattering increases, and also wavelength dependent. DPF value can be estimated when absorption and scattering coefficients are known 

Ref) J. of Biomedical Optics, 18(10), 105004 (2013).          https://doi.org/10.1117/1.JBO.18.10.105004

3. As you can see in Fig. 5-6, all the hemodynamic signals start at zero and shows the relative changes of oxy-, deoxy- and total hemoglobin concentration. Since CW-NIRS does not provide absolute values of hemoglobin concentration, a physiological intervention is required such as apomorphine administration in this study, to induce a hemodynamic change which can be observed by CW-NIRS. In a clinical setting, audiovisual sexual stimulation can replace the administration of apomorphine.

[Minor changes]

Point 1: In the title “rat model” sounds like the final aim of this study is to investigate sexual arousal in rats. Please, modify the title.

Response 1

The title has been revised from “Vaginal Hemodynamic Change During Sexual Arousal in a Rat Model Using a Diffuse Optical Probe” to Design of Optical Probe to Monitor Vaginal Hemodynamics during Sexual Arousal”

Page. 1, line 2 – Page. 1, line 3

Point 2: DPF” abbreviation on Figures 5,6 should be decrypted

Response 2

Page 5, line 146, Figure. 5

The DPF abbreviation has been decrypted as below.

“ DPF: Differential pathlength factor ”

Page 5, line 153, Figure. 6

The DPF abbreviation has been decrypted as below.

“ DPF: Differential pathlength factor ”

Round 2

Reviewer 2 Report

The authors have accurately answered all addressed questions and the quality of paper has been improved. I’d like to recommend the paper for publication, however, some minor changes should be applied before it.

1. In the motivation of the study mostly old papers have been referenced [5,6,7,8,10,12]. It would be beneficial if the authors include the latest publications (<5 years old), at least conference proceedings.

2. Line 59: “It is also known to be sensitive to microvessels rather than large blood vessels..”. It is not entirely correct, because NIRS is sensitive to the blood content for both large vessels and microvessels, because it is sensitive to the overall tissue chromophores. However, if microvessels are located more closely to the tissue surface, they providing higher yield in the NIRS signal changes, resulting in the overall higher sensitivity to the blood content in the microvessels rather than in large vessels.

Author Response

Point 1In the motivation of the study mostly old papers have been referenced [5,6,7,8,10,12]. It would be beneficial if the authors include the latest publications (<5 years old), at least conference proceedings.

Response 1Line 232-244, Line 247-249

References were replaced with the ones published within the last 5 years.

The revised ones have been highlighted by yellow.

Point 2Line 59: “It is also known to be sensitive to microvessels rather than large blood vessels..”. It is not entirely correct, because NIRS is sensitive to the blood content for both large vessels and microvessels, because it is sensitive to the overall tissue chromophores. However, if microvessels are located more closely to the tissue surface, they providing higher yield in the NIRS signal changes, resulting in the overall higher sensitivity to the blood content in the microvessels rather than in large vessels.

Response 2Line 59-61

Thank you for your comment.
We agree that NIRS is sensitive to both large and micro blood vessels and the current description can confuse the readers.

Therefore, we revised the sentence as follows.

“It is also known to be sensitive to microvessels which are widely distributed in the vaginal wall, and thus it fits well for the purpose of monitoring microvascular hemodynamics in the vaginal wall compared to LDF [12]."